# $\sigma$-ZERO: GRADIENT-BASED OPTIMIZATION OF $\ell_0$-NORM ADVERSARIAL EXAMPLES

**Antonio Emanuele Cinà**[1]    **Francesco Villani**[1]    **Maura Pintor**[2]    **Lea Schönherr**[3]
**Battista Biggio**[2]    **Marcello Pelillo**[4]

[1]Department of Computer Science, Bioengineering, Robotics and Systems, University of Genoa, Italy

[2]Department of Electrical and Electronic Engineering, University of Cagliari, Italy

[3]CISPA Helmholtz Center for Information Security, Germany

[4]Department of Environmental Sciences, Informatics and Statistics, Ca' Foscari University of Venice, Italy

antonio.cina@unige.it   francesco.villani@edu.unige.it   maura.pintor@unica.it
schoenherr@cispa.de   battista.biggio@unica.it   pelillo@unive.it

## ABSTRACT

Evaluating the adversarial robustness of deep networks to gradient-based attacks is challenging. While most attacks consider $\ell_2$- and $\ell_\infty$-norm constraints to craft input perturbations, only a few investigate sparse $\ell_1$- and $\ell_0$-norm attacks. In particular, $\ell_0$-norm attacks remain the least studied due to the inherent complexity of optimizing over a non-convex and non-differentiable constraint. However, evaluating adversarial robustness under these attacks could reveal weaknesses otherwise left untested with more conventional $\ell_2$- and $\ell_\infty$-norm attacks. In this work, we propose a novel $\ell_0$-norm attack, called $\sigma$-zero, which leverages a differentiable approximation of the $\ell_0$ norm to facilitate gradient-based optimization, and an adaptive projection operator to dynamically adjust the trade-off between loss minimization and perturbation sparsity. Extensive evaluations using MNIST, CIFAR10, and ImageNet datasets, involving robust and non-robust models, show that $\sigma$-zero finds minimum $\ell_0$-norm adversarial examples without requiring any time-consuming hyperparameter tuning, and that it outperforms all competing sparse attacks in terms of success rate, perturbation size, and efficiency.

## 1 INTRODUCTION

Early research has revealed that machine learning models are fooled by adversarial examples, i.e., slightly-perturbed inputs optimized to cause misclassifications (Biggio et al., 2013; Szegedy et al., 2014). The discovery of this phenomenon has, in turn, demanded a more careful evaluation of the robustness of such models, especially when deployed in security-sensitive and safety-critical applications. Most of the gradient-based attacks proposed to evaluate the adversarial robustness of Deep Neural Networks (DNNs) optimize adversarial examples under different $\ell_p$-norm constraints. In particular, while convex $\ell_1$, $\ell_2$, and $\ell_\infty$ norms have been widely studied (Chen et al., 2018; Croce & Hein, 2021), only a few $\ell_0$-norm attacks have been considered to date. The main reason is that finding minimum $\ell_0$-norm solutions is known to be an NP-hard problem (Davis et al., 1997), and thus ad-hoc approximations must be adopted to overcome issues related to the non-convexity and non-differentiability of such (pseudo) norm. Although this is a challenging task, attacks based on the $\ell_0$ norm have the potential to uncover issues in DNNs that may not be evident when considering other attacks (Carlini & Wagner, 2017b; Croce & Hein, 2021). In particular, $\ell_0$-norm attacks, known to perturb a minimal fraction of input values, can be used to determine the most sensitive characteristics that influence the model's decision-making process, offering a different and relevant threat model to benchmark existing defenses and a different understanding of the model's inner workings.

Unfortunately, current $\ell_0$-norm attacks exhibit a largely suboptimal trade-off between their success rate and efficiency, i.e., they are either accurate but slow or fast but inaccurate. In particular, the accurate ones use complex projections and advanced initialization strategies (e.g., adversarial

---

Code is available at https://github.com/sigma0-advx/sigma-zero.

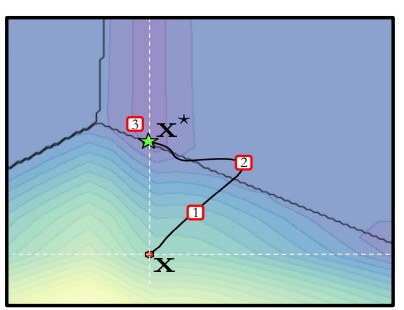 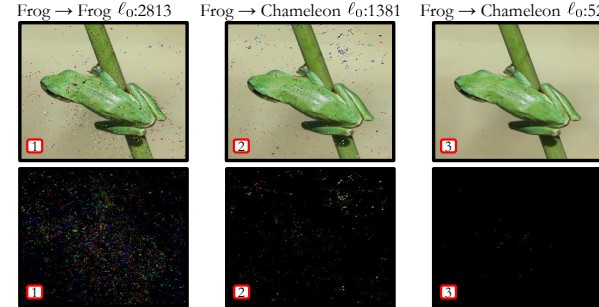

Figure 1: The leftmost plot shows the execution of $\sigma$-zero on a two-dimensional problem. The initial point $\mathbf{x}$ (*red dot*) is updated via gradient descent to find the adversarial example $\mathbf{x}^\star$ (*green star*) while minimizing the number of perturbed features (i.e., the $\ell_0$ norm of the perturbation). The gray lines surrounding $\mathbf{x}$ demarcate regions where the $\ell_0$ norm is minimized. The rightmost plot shows the adversarial images (*top row*) and the corresponding perturbations (*bottom row*) found by $\sigma$-zero during the three steps highlighted in the leftmost plot, along with their prediction and $\ell_0$ norm.

initialization) to find smaller input perturbations but suffer from time or memory limitations, hindering their scalability to larger networks or high-dimensional data (Brendel et al., 2019a; Césaire et al., 2021). Other attacks execute faster, but their returned solution is typically less accurate and largely suboptimal (Matyasko & Chau, 2021; Pintor et al., 2021). This results in overestimating adversarial robustness and, in turn, contributes to spreading a *false sense of security*, hindering the development of effective defense mechanisms (Carlini et al., 2019; Pintor et al., 2022). Developing a reliable, scalable, and compelling method to assess the robustness of DNN models against sparse perturbations with minimum $\ell_0$ norm remains thus a relevant and challenging open problem.

In this work, we propose a novel $\ell_0$-norm attack, named $\sigma$-zero, which iteratively promotes the sparsity of the adversarial perturbation by minimizing its $\ell_0$ norm (see Figure 1 and Sect. 2). To overcome the limitations of previous approaches, our attack leverages two main technical contributions: (i) a smooth, differentiable approximation of the $\ell_0$ norm to enable the minimization of the attack loss via gradient descent; and (ii) an adaptive projection operator that dynamically increases sparsity to further reduce the perturbation size while keeping the perturbed sample in the adversarial region.

Our experiments (Sect. 3) provide compelling evidence of the remarkable performance of $\sigma$-zero. We evaluate it on 3 well-known benchmark datasets (i.e., MNIST, CIFAR10, and ImageNet), using 22 different models from Robustbench (Croce et al., 2021) and the corresponding official repositories. We compare the performance of $\sigma$-zero against more than 10 competing attacks, totaling almost 450 different comparisons. Our analysis shows that $\sigma$-zero outperforms state-of-the-art attacks in terms of both attack success rate and perturbation size (lower $\ell_0$ norm), while being also significantly faster (i.e., requiring fewer queries and lower runtime). Our attack also provides some additional advantages: (i) it does not require any sophisticated, time-consuming hyperparameter tuning; (ii) it does not require being initialized from an adversarial input; (iii) it is less likely to fail, i.e., it consistently achieves an attack success rate of 100% for sufficiently-large perturbation budgets, thereby enabling more reliable robustness evaluations (Carlini et al., 2019). We thus believe that $\sigma$-zero will foster significant advancements in the development of better robustness evaluation tools and more robust models against sparse attacks. We conclude the paper by discussing related work (Sect. 4), along with the main contributions and future research directions (Sect. 5).

## 2 $\sigma$-ZERO: MINIMUM $\ell_0$-NORM ATTACKS

We present here $\sigma$-zero, a gradient-based attack that finds minimum $\ell_0$-norm adversarial examples.

**Threat Model.** We assume that the attacker has complete access to the target model, including its architecture and trained parameters, and exploits its gradient for staging white-box untargeted attacks (Carlini & Wagner, 2017b; Biggio & Roli, 2018). This setting is useful for worst-case evaluation of the adversarial robustness of DNNs, providing an empirical assessment of the performance degradation that may be incurred under attack. Note that this is the standard setting adopted

in previous work for gradient-based adversarial robustness evaluations (Carlini & Wagner, 2017b; Brendel et al., 2019b; Croce et al., 2021; Pintor et al., 2021).

**Problem Formulation.** In this work, we seek untargeted minimum $\ell_0$-norm adversarial perturbations that steer the model's decision towards misclassification (Carlini & Wagner, 2017b). To this end, let $\mathbf{x} \in \mathcal{X} = [0,1]^d$ be a $d$-dimensional input sample, $y \in \mathcal{Y} = \{1, \ldots, l\}$ its associated true label, and $f : \mathcal{X} \times \Theta \mapsto \mathcal{Y}$ the target model, parameterized by $\boldsymbol{\theta} \in \Theta$. While $f$ outputs the predicted label, we will also use $f_k$ to denote the continuous-valued output (logit) for class $k \in \mathcal{Y}$. The goal of our attack is to find the minimum $\ell_0$-norm adversarial perturbation $\boldsymbol{\delta}^\star$ such that the corresponding adversarial example $\mathbf{x}^\star = \mathbf{x} + \boldsymbol{\delta}^\star$ is misclassified by $f$. This can be formalized as:

$$\boldsymbol{\delta}^\star \in \arg\min_{\boldsymbol{\delta}} \quad \|\boldsymbol{\delta}\|_0 \,, \tag{1}$$

$$\text{s.t.} \quad f(\mathbf{x} + \boldsymbol{\delta}, \boldsymbol{\theta}) \neq y \,, \tag{2}$$

$$\mathbf{x} + \boldsymbol{\delta} \in [0,1]^d \,, \tag{3}$$

where $\| \cdot \|_0$ denotes the $\ell_0$ norm, which counts the number of non-zero components. The hard constraint in Eq. (2) ensures that the perturbation $\boldsymbol{\delta}$ is valid only if the target model $f$ misclassifies the perturbed sample $\mathbf{x} + \boldsymbol{\delta}$, while the box constraint in Eq. (3) ensures that the perturbed sample lies in $[0,1]^d$.[1] Since the problem in Eqs. (1)-(3) can not be solved directly, we reformulate it as:

$$\boldsymbol{\delta}^\star \in \arg\min_{\boldsymbol{\delta}} \quad \mathcal{L}(\mathbf{x} + \boldsymbol{\delta}, y, \boldsymbol{\theta}) + \frac{1}{d}\hat{\ell}_0(\boldsymbol{\delta}) \tag{4}$$

$$\text{s.t.} \quad \mathbf{x} + \boldsymbol{\delta} \in [0,1]^d \,, \tag{5}$$

where we use a differentiable approximation $\hat{\ell}_0(\boldsymbol{\delta})$ instead of $||\boldsymbol{\delta}||_0$, and normalize it with respect to the number of features $d$ to ensure that its value is within the interval $[0,1]$. The loss $\mathcal{L}$ is defined as:

$$\mathcal{L}(\mathbf{x}, y, \boldsymbol{\theta}) = \max\left( f_y(\mathbf{x}, \boldsymbol{\theta}) - \max_{k \neq y} f_k(\mathbf{x}, \boldsymbol{\theta}), 0 \right) + \mathbb{I}(f(\mathbf{x}, \boldsymbol{\theta}) = y) \,. \tag{6}$$

The first term in $\mathcal{L}$ represents the logit difference, which is positive when the sample is correctly assigned to the true class $y$, and clipped to zero when it is misclassified (Carlini & Wagner, 2017b). The second term merely adds 1 to the loss if the sample is correctly classified.[2] This ensures that $\mathcal{L} = 0$ only when an adversarial example is found and $\mathcal{L} \geq 1$ otherwise. In practice, when minimizing the objective in Eq. (4), this loss term induces an *alternate* optimization process between minimizing the loss function itself (to find an adversarial example) and minimizing the $\ell_0$-norm of the adversarial perturbation (when an adversarial example is found). It is also worth remarking that, conversely to the objective function proposed by Carlini & Wagner (2017b), our objective does not require tuning any trade-off hyperparameters to balance between minimizing the loss and reducing the perturbation size, thereby avoiding a computationally expensive line search for each input sample.

$\ell_0$**-norm Approximation.** Besides the formalization of the attack objective, one of the main technical advantages of $\sigma\text{-zero}$ is the smooth, differentiable approximation of the $\ell_0$ norm, thereby enabling the use of gradient-based optimization. To this end, we first note that the $\ell_0$-norm of a vector can be rewritten as $\|\mathbf{x}\|_0 = \sum_{i=1}^{d} \text{sign}(x_i)^2$, and then approximate the *sign* function as $\text{sign}(x_i) \approx x_i/\sqrt{x_i^2 + \sigma}$, where $\sigma > 0$ is a smoothing hyperparameter that makes the approximation sharper as $\sigma \to 0$. This, in turn, yields the following smooth approximation of the $\ell_0$ norm:

$$\hat{\ell}_0(\mathbf{x}, \sigma) = \sum_{i=1}^{d} \frac{x_i^2}{x_i^2 + \sigma}, \sigma > 0, \quad \hat{\ell}_0(\mathbf{x}, \sigma) \in [0, d] \,. \tag{7}$$

**Adaptive Projection $\Pi_\tau$.** The considered $\ell_0$-norm approximation allows optimizing Eq. (4) via gradient descent. However, using such a smooth approximation tends to promote solutions that are not fully sparse, i.e., with many components that are very close to zero but not exactly equal to zero, thereby yielding inflated $\ell_0$-norm values. To overcome this issue, we introduce an adaptive projection operator $\Pi_\tau$ that sets to zero the components with a perturbation intensity lower than a given *sparsity*

---

[1]Note that, when the source point $\mathbf{x}$ is already misclassified by $f$, the solution is simply $\boldsymbol{\delta}^\star = \mathbf{0}$.

[2]While a sigmoid approximation may be adopted to overcome the non-differentiability of the $\mathbb{I}$ term at the decision boundary, we simply set its gradient to zero *everywhere*, without any impact on the experimental results.

---

**Algorithm 1** $\sigma$-zero Attack Algorithm.

---

**Input:** $\mathbf{x} \in [0, 1]^d$, the input sample; y, the true class label; $\boldsymbol{\theta}$, the target model; N, the number of
iterations; $\eta_0 = 1.0$, the initial step size; $\sigma = 10^{-3}$, the $\ell_0$-norm smoothing hyperparameter;
$\tau_0 = 0.3$, the initial sparsity threshold; $t = 0.01$, the sparsity threshold adjustment factor.

**Output:** $\mathbf{x}^\star$, the minimum $\ell_0$-norm adversarial example.

1   $\boldsymbol{\delta} \leftarrow \mathbf{0}; \quad \boldsymbol{\delta}^\star \leftarrow \infty; \quad \tau \leftarrow \tau_0; \quad \eta \leftarrow \eta_0$
2   **for** $i$ in $1, \ldots, N$ **do**
3     $\nabla \mathbf{g} \leftarrow \nabla_{\boldsymbol{\delta}}[\mathcal{L}(\mathbf{x} + \boldsymbol{\delta}, y, \boldsymbol{\theta}) + \frac{1}{d}\hat{\ell}_0(\boldsymbol{\delta}, \sigma)]$       ▷ Gradient Descent for Eq. (4).
4     $\nabla \mathbf{g} \leftarrow \nabla \mathbf{g}/\|\nabla \mathbf{g}\|_\infty$                   ▷ Gradient Normalization.
5     $\boldsymbol{\delta} \leftarrow \text{clip}(\mathbf{x} + [\boldsymbol{\delta} - \eta \cdot \nabla \mathbf{g}]) - \mathbf{x}$           ▷ Box Constraints.
6     $\boldsymbol{\delta} \leftarrow \Pi_\tau(\boldsymbol{\delta})$                   ▷ Adaptive Projection Operator.
7     $\eta = \text{cosine\_annealing}(\eta_0, i)$           ▷ Learning Rate Decay.
8     **if** $\mathcal{L}(\mathbf{x} + \boldsymbol{\delta}, y, \boldsymbol{\theta}) \leq 0$: $\tau + = t \cdot \eta$, **else** $\tau - = t \cdot \eta$    ▷ Adaptive Adjustment for $\tau$.
9     **if** $\mathcal{L}(\mathbf{x} + \boldsymbol{\delta}, y, \boldsymbol{\theta}) \leq 0 \ \wedge \ \|\boldsymbol{\delta}\|_0 < \|\boldsymbol{\delta}^\star\|_0$: $\boldsymbol{\delta}^\star \leftarrow \boldsymbol{\delta}$
10   **end**
11   **if** $\mathcal{L}(\mathbf{x} + \boldsymbol{\delta}^\star, y, \boldsymbol{\theta}) > 0$: $\boldsymbol{\delta}^\star \leftarrow \infty$
12   **return** $\mathbf{x}^\star \leftarrow \mathbf{x} + \boldsymbol{\delta}^\star$

---

*threshold* $\tau$ in each iteration. The sparsity threshold $\tau$ is initialized with a starting value $\tau_0$ and then dynamically adjusted for each sample during each iteration; in particular, it is increased to find sparser perturbations when the current sample is already adversarial, while it is decreased otherwise. The updates to $\tau$ are proportional to the step size and follow its annealing strategy, as detailed below.

**Solution Algorithm.** Our attack, given as Algorithm 1, solves the problem in Eqs. (4)-(5) via a fast and memory-efficient gradient-based optimization. After initializing the adversarial perturbation $\boldsymbol{\delta} = \mathbf{0}$ (line 1), it computes the gradient of the objective in Eq. (4) with respect to $\boldsymbol{\delta}$ (line 3). The gradient is then normalized such that its largest components (in absolute value) equal $\pm 1$ (line 4). This stabilizes the optimization by making the update independent from the gradient size, and also makes the selection of the step size independent from the input dimensionality (Rony et al., 2018; Pintor et al., 2021). We then update $\boldsymbol{\delta}$ to minimize the objective via gradient descent while also enforcing the box constraints in Eq. (5) through the usage of the clip operator (line 5). We increase sparsity in $\boldsymbol{\delta}$ by zeroing all components lower than the current sparsity threshold $\tau$ (line 6), as discussed in the previous paragraph. We then decrease the step size $\eta$ via cosine annealing (line 7), as suggested by Rony et al. (2018); Pintor et al. (2021), and adjust the sparsity threshold $\tau$ accordingly (line 8). In particular, if the current sample is adversarial, we increase $\tau$ by $t \cdot \eta$ to promote sparser perturbations; otherwise, we decrease $\tau$ by the same amount to promote the minimization of $\mathcal{L}$. The above process is repeated for $N$ iterations while keeping track of the best solution found, i.e., the adversarial perturbation $\boldsymbol{\delta}^\star$ with the lowest $\ell_0$ norm (line 9). If no adversarial example is found, the algorithm sets $\boldsymbol{\delta}^\star = \infty$ (line 11). It terminates by returning $\mathbf{x}^\star = \mathbf{x} + \boldsymbol{\delta}^\star$ (line 12).

**Remarks.** To summarize, the main contributions behind $\sigma$-zero are: (i) the use of a smooth $\ell_0$-norm approximation, along with the definition of an appropriate objective (Eq. 4), to enable optimizing $\ell_0$-norm adversarial examples via gradient descent; and (ii) the introduction of an adaptive projection operator to further improve sparsity during the optimization. Our algorithm leverages also common strategies like gradient normalization and step size annealing to speed up convergence. As reported by our experiments, $\sigma$-zero provides a more effective and efficient $\ell_0$-norm attack that (i) is robust to different hyperparameter choices; (ii) does not require any adversarial initialization; and (iii) enables more reliable robustness evaluations, being able to find adversarial examples also when the competing attacks may fail (Carlini et al., 2019; Pintor et al., 2022).

## 3   EXPERIMENTS

We report here an extensive experimental evaluation comparing $\sigma$-zero against 11 state-of-the-art sparse attacks, including both $\ell_0$- and $\ell_1$-norm attacks. We test all attacks using different settings on 18 distinct models and 3 different datasets, yielding almost 450 different comparisons in total.

## 3.1 EXPERIMENTAL SETUP

**Datasets.** We consider the three most popular datasets used for benchmarking adversarial robustness: MNIST (LeCun & Cortes, 2005), CIFAR-10 (Krizhevsky, 2009) and ImageNet (Krizhevsky et al., 2012). To evaluate the attack performance, we use the entire test set for MNIST and CIFAR-10 (with a batch size of 32), and a subset of 1000 test samples for ImageNet (with a batch size of 16).

**Models.** We use a selection of both baseline and robust models to evaluate the attacks under different conditions. We evaluate $\sigma$-zero on a vast set of models to ensure its broad effectiveness and expose vulnerabilities that may not be revealed by other attacks (Croce & Hein, 2021). For the MNIST dataset, we consider two adversarially trained convolutional neural network (CNN) models by Rony et al. (2021), i.e., CNN-DDN and CNN-Trades. These models have been trained to be robust to both $\ell_2$ and $\ell_\infty$ adversarial attacks. We denote them M1 and M2, respectively. For the CIFAR-10 and ImageNet datasets, we employ state-of-the-art robust models from RobustBench (Croce et al., 2021) and the paper's official repositories. For CIFAR-10, we adopt ten models, denoted as C1-C12. C1 (Carmon et al., 2019) and C2 (Augustin et al., 2020) combine training data augmentation with adversarial training to improve robustness to $\ell_\infty$ and $\ell_2$ attacks. C3 (Croce & Hein, 2021) and C4 (Jiang et al., 2023) are $\ell_1$ robust models. C5 (Croce et al., 2021) is a non-robust WideResNet-28-10 model. C6 (Gowal et al., 2021) uses generative models to artificially augment the original training set and improve adversarial robustness to generic $\ell_p$-norm attacks. C7 (Engstrom et al., 2019) is an adversarially trained model that is robust against $\ell_2$-norm attacks. C8 (Chen et al., 2020) is a robust ensemble model. C9 (Xu et al., 2023) is a recently proposed adversarial training defense robust to $\ell_2$ attacks. C10 (Addepalli et al., 2022) enforces diversity during data augmentation and combines it with adversarial training. Lastly, C11 (Zhong et al., 2024) and C12 (Zhong et al., 2024) are two adversarially trained models robust against $\ell_0$-norm adversarial perturbations. For ImageNet, we consider a pretrained ResNet-18 denoted with I1 (He et al., 2015), and five robust models to $\ell_\infty$-attacks, denoted with I2 (Engstrom et al., 2019), I3 (Hendrycks et al., 2021), I4 (Debenedetti et al., 2023), I5 (Wong et al., 2020), and I6 (Salman et al., 2020). Lastly, in the appendix, we present two $\ell_0$-robust models, C11 (Zhong et al., 2024) and C12 (Zhong et al., 2024), for CIFAR-10, along with two large $\ell_\infty$-robust models, I7 (Peng et al., 2023) and I8 (Mo et al., 2022), for ImageNet.

**Attacks.** We compare $\sigma$-zero against the following state-of-the-art minimum-norm attacks, in their $\ell_0$-norm variants: the Voting Folded Gaussian Attack (VFGA) attack (Césaire et al., 2021), the Primal-Dual Proximal Gradient Descent (PDPGD) attack (Matyasko & Chau, 2021), the Brendel & Bethge (BB) attack (Brendel et al., 2019a), including also its variant with adversarial initialization (BBadv),[3] and the Fast Minimum Norm (FMN) attack (Pintor et al., 2021). We also consider two state-of-the-art $\ell_1$-norm attacks as additional baselines, i.e., the Elastic-Net (EAD) attack (Chen et al., 2018) and SparseFool (SF) by Modas et al. (2019). All attacks are set to manipulate the input values independently; e.g., for CIFAR-10, the number of modifiable inputs is $3 \times 32 \times 32 = 3072$.

**Hyperparameters.** We run our experiments using the default hyperparameters from the original implementations provided in the authors' repositories, *AdversarialLib* (Rony & Ben Ayed) and *Foolbox* (Rauber et al., 2017). We set the maximum number of iterations to $N = 1000$ to ensure that all attacks reach convergence (Pintor et al., 2022).[4] For $\sigma$-zero, we set $\eta_0 = 1$, $\tau_0 = 0.3$, $t = 0.01$, and $\sigma = 10^{-3}$, and keep the same configuration for all models and datasets.[5]

**Evaluation Metrics.** For each attack, we report the Attack Success Rate (ASR) at different values of $k$, denoted with $\text{ASR}_k$, i.e., the fraction of successful attacks for which $\|\boldsymbol{\delta}^\star\|_0 \leq k$, and the median value of $\|\boldsymbol{\delta}^\star\|_0$ over the test samples, denoted with $\tilde{\ell}_0$.[6] We compare the computational effort of each attack considering the mean runtime (**s**) (per sample), the mean number of queries (**q**) (i.e., the total number of forwards and backwards required to perform the attack, divided by the number of samples), and the Video Random Access Memory (VRAM) consumed by the Graphics Processing Unit (GPU). We measure the runtime on a workstation with an NVIDIA A100 Tensor Core GPU (40 GB memory) and two Intel® Xeo® Gold 6238R processors. We evaluate memory consumption as the maximum VRAM used among all batches, representing the minimum requirement to run without failure.

---

[3]We utilize the Foolbox DatasetAttack (Foolbox, 2017) for adversarial initialization.

[4]Additional results using only $N = 100$ steps are reported in Appendix B.1.

[5]To show that no specific hyperparameter tuning is required, additional results are reported in Appendix A.2.

[6]If no adversarial example is found for a given $\mathbf{x}$, we set $\|\boldsymbol{\delta}^\star\|_0 = \infty$, as done by Brendel et al. (2019a).

Table 1: Minimum-norm comparison results on MNIST, CIFAR10 and ImageNet with $N = 1000$. For each attack and model (M), we report ASR at $k = 24, 50, \infty$, median perturbation size $\tilde{\ell}_0$, mean runtime $s$ (in seconds), mean number of queries $q$ (in thousands), and maximum VRAM usage (in GB). When VFGA exceeds the VRAM limit, we re-run it using a smaller batch size, increasing its runtime $t$. We denote those cases with the symbol '$\star$'. Remaining models in Appendix B, Table 6.

| Attack | M | $ASR_{24}$ | $ASR_{50}$ | $ASR_{\infty}$ | $\tilde{\ell}_0$ | s | q | VRAM | M | $ASR_{24}$ | $ASR_{50}$ | $ASR_{\infty}$ | $\tilde{\ell}_0$ | s | q | VRAM |
|---|---|---|---|---|---|---|---|---|---|---|---|---|---|---|---|---|
| | | | | | | | | MNIST | | | | | | | | |
| SF | | 6.66 | 6.76 | 96.98 | 469 | 1.07 | 0.18 | 0.06 | | 1.03 | 1.21 | 91.68 | 463 | 2.87 | 0.86 | 0.07 |
| EAD | | 3.83 | 53.66 | 100.0 | 49 | 0.47 | 6.28 | 0.05 | | 2.13 | 55.57 | 100.0 | 48 | 0.50 | 6.73 | 0.05 |
| PDPGD | | 26.77 | 74.08 | 100.0 | 38 | 0.23 | 2.00 | 0.04 | | 16.91 | 66.30 | 100.0 | 42 | 0.23 | 2.00 | 0.04 |
| VFGA | M1 | 43.58 | 82.42 | 99.98 | 27 | 0.05 | 0.77 | 0.21 | M2 | 5.00 | 39.33 | 99.95 | 57 | 0.05 | 1.33 | 0.21 |
| FMN | | 35.90 | 93.74 | 100.0 | 29 | 0.21 | 2.00 | 0.04 | | 50.74 | 91.84 | 99.41 | 24 | 0.22 | 2.00 | 0.04 |
| BB | | 71.23 | 97.86 | 100.0 | 18 | 0.90 | 2.99 | 0.05 | | 56.53 | 91.62 | 100.0 | 18 | 0.74 | 3.71 | 0.05 |
| BBadv | | 67.06 | 91.23 | 100.0 | 19 | 0.77 | 2.01 | 0.07 | | 29.17 | 40.88 | 100.0 | 89 | 0.71 | 2.01 | 0.07 |
| σ-zero | | **83.79** | **99.98** | 100.0 | **16** | 0.31 | 2.00 | 0.04 | | **98.03** | **100.0** | 100.0 | **9** | 0.31 | 2.00 | 0.04 |
| | | | | | | | | CIFAR-10 | | | | | | | | |
| SF | | 18.71 | 18.77 | 56.39 | 3072 | 11.31 | 1.40 | 1.62 | | 20.46 | 24.36 | 58.29 | 3072 | 1.63 | 0.48 | 0.66 |
| EAD | | 16.32 | 30.38 | 100.0 | 90 | 1.92 | 5.70 | 1.47 | | 13.01 | 13.23 | 100.0 | 800 | 0.94 | 4.89 | 0.65 |
| PDPGD | | 26.84 | 42.50 | 100.0 | 63 | 0.64 | 2.00 | 1.32 | | 22.30 | 35.13 | 100.0 | 75 | 0.41 | 2.00 | 0.59 |
| VFGA | C1 | 51.06 | 75.37 | 99.92 | 24 | 0.59 | 0.78 | 11.71 | C3 | 28.47 | 49.98 | 99.72 | 51 | 0.32 | 1.25 | 4.44 |
| FMN | | 48.89 | 74.70 | 100.0 | 26 | 0.59 | 2.00 | 1.31 | | 27.45 | 48.87 | 100.0 | 52 | 0.24 | 2.00 | 0.60 |
| BB | | 13.27 | 14.24 | 14.70 | $\infty$ | 0.63 | 2.05 | 1.47 | | 16.88 | 22.91 | 27.64 | $\infty$ | 1.04 | 2.25 | 0.65 |
| BBadv | | 65.96 | 90.57 | 100.0 | 16 | 4.68 | 2.01 | 1.64 | | 36.47 | 72.43 | 100.0 | 34 | 5.28 | 2.01 | 0.64 |
| σ-zero | | **76.53** | **95.38** | 100.0 | **11** | 0.73 | 2.00 | 1.53 | | **38.60** | **73.02** | 100.0 | 32 | 0.43 | 2.00 | 0.71 |
| SF | | 19.66 | 21.22 | 98.74 | 3070 | 3.62 | 0.46 | 1.90 | | 31.76 | 43.07 | 91.14 | 69 | 4.32 | 1.49 | 0.66 |
| EAD | | 9.73 | 11.42 | 100.0 | 360 | 2.53 | 5.62 | 1.89 | | 24.21 | 24.78 | 100.0 | 768 | 1.04 | 4.99 | 0.65 |
| PDPGD | | 28.02 | 45.15 | 100.0 | 55 | 1.12 | 2.00 | 1.8 | | 26.89 | 42.38 | 100.0 | 66 | 0.40 | 2.00 | 0.60 |
| VFGA | C2 | 39.58 | 66.50 | 99.62 | 34 | 0.48 | 0.94 | 16.53 | C4 | 46.71 | 69.47 | 99.83 | 28 | 0.25 | 0.82 | 4.22 |
| FMN | | 39.30 | 71.70 | 100.0 | 33 | 1.08 | 2.00 | 1.8 | | 43.06 | 62.96 | 100.0 | 34 | 0.35 | 2.00 | 0.59 |
| BB | | 38.73 | 56.78 | 58.64 | 33 | 2.31 | 2.89 | 1.89 | | 25.95 | 27.98 | 29.50 | $\infty$ | 0.54 | 2.09 | 0.65 |
| BBadv | | 70.07 | 96.31 | 100.0 | 17 | 3.92 | 2.01 | 1.99 | | 53.17 | 82.46 | 100.0 | 22 | 3.03 | 2.01 | 0.65 |
| σ-zero | | **74.63** | **97.55** | 100.0 | **15** | 1.41 | 2.00 | 1.92 | | **55.42** | **82.92** | 100.0 | **20** | 0.42 | 2.00 | 0.72 |
| | | | | | | | | ImageNet | | | | | | | | |
| EAD | | 35.4 | 36.3 | 100.0 | 460 | 4.13 | 2.69 | 0.46 | | 27.0 | 28.4 | 100.0 | 981 | 19.25 | 5.49 | 1.41 |
| VFGA | | 57.9 | 72.5 | 99.9 | 14 | 1.22* | 1.08 | >40 | | 46.7 | 59.5 | 97.9 | 31 | 6.93* | 1.98 | >40 |
| FMN | I1 | 62.6 | 81.0 | 100.0 | 12 | 0.73 | 2.00 | 0.66 | I3 | 49.1 | 67.7 | 100.0 | 25 | 1.98 | 2.00 | 2.30 |
| BBadv | | 77.5 | 93.2 | 100.0 | 7 | 231.67 | 2.01 | 0.72 | | 64.7 | 85.5 | 100.0 | 14 | 205.11 | 2.01 | 2.41 |
| σ-zero | | **82.6** | **95.9** | 100.0 | **5** | 1.18 | 2.00 | 0.84 | | **66.7** | **86.9** | 100.0 | **13** | 2.76 | 2.00 | 2.52 |
| EAD | | 46.8 | 51.0 | 100.0 | 42 | 18.10 | 5.45 | 1.42 | | 32.8 | 33.5 | 100.0 | 572 | 11.43 | 5.34 | 1.68 |
| VFGA | | 54.7 | 63.4 | 96.7 | 12 | 8.21* | 2.35 | >40 | | 40.0 | 46.5 | 95.5 | 66 | 33.88* | 3.97 | >40 |
| FMN | I2 | 57.8 | 67.0 | 100.0 | 9 | 1.97 | 2.00 | 2.30 | I4 | 40.3 | 47.2 | 100.0 | 58 | 4.28 | 2.00 | 2.97 |
| BBadv | | 71.0 | 82.3 | 100 | 4 | 182.65 | 2.01 | 2.40 | | 46.8 | 59..8 | 100.0 | 31 | 178.06 | 2.01 | 3.07 |
| σ-zero | | **76.9** | **87.4** | 100.0 | **3** | 2.75 | 2.00 | 2.52 | | **50.7** | **65.1** | 100.0 | **23** | 5.72 | 2.00 | 3.20 |

## 3.2 EXPERIMENTAL RESULTS

We report the success rate and computational effort metrics of σ-zero against minimum-norm attacks in Table 1 and fixed-budget attacks in Table 3-4. In these tables, we consider the most robust models for each dataset, and we provide the remaining results in Appendix B. Finally, for ImageNet, we narrow our analysis to EAD, FMN, BBadv, and VFGA minimum-norm attacks, as they surpass competing attacks on MNIST and CIFAR-10 in terms of ASR, perturbation size, or execution time.

**Effectiveness.** The median values of $||\boldsymbol{\delta}^{\star}||_0$, denoted as $\tilde{\ell}_0$, and the ASRs are reported in Table 1 for all models and datasets. To facilitate comparison, the attacks are sorted from the least to the most effective, on average. In all dataset-model configurations, σ-zero significantly outperforms all the considered attacks. Taking the best-performing attack among the fastest competitors as a reference (i.e., FMN), σ-zero is able to find smaller perturbations and higher ASRs in all configurations. In particular, on CIFAR-10, σ-zero reduces the median number of manipulated features from 52 to 32 against the most robust model (C3), with an average reduction of $49\%$ across all models. On ImageNet, this improvement is even more pronounced, with a reduction of up to $58\%$. In the best case (I4), the median $||\boldsymbol{\delta}^{\star}||_0$ is reduced from 58 to 23, and in the worst case (I2), from 9 to 3. Alternatively, the most competitive attack in finding small perturbations is BBadv, which is significantly slower and requires starting from an already-adversarial input. The $ASR_{\infty}$ of BB

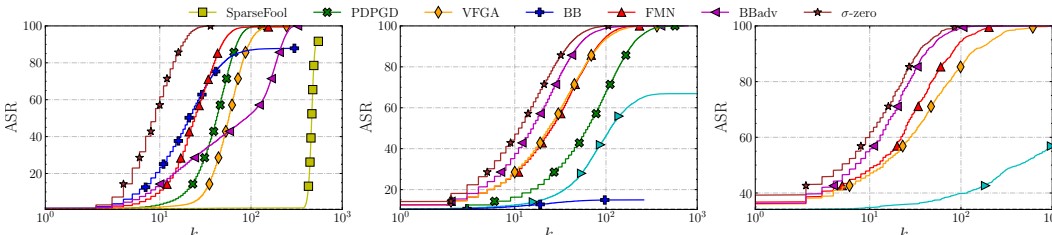

Figure 2: Robustness evaluation curves (ASR vs. perturbation budget $k$) for M2 on MNIST (*left*), C1 on CIFAR-10 (*middle*), and I1 on ImageNet (*right*).

(i.e., without adversarial initialization) indeed decreases with increasing input dimensionality (e.g., CIFAR-10). This occurs because BB often stops unexpectedly before reaching the specified number of steps due to initialization failures; in particular, Table 1 shows that the median perturbation size found by BB is sometimes $\infty$, as its $ASR_\infty$ is lower than $50\%$. Although BBadv does not suffer from the same issue, as it leverages adversarial initialization, it is still outperformed by $\sigma\text{-zero}$. Specifically, $\sigma\text{-zero}$ reduces the $\ell_0$ norm of the adversarial examples from 16 to 11 in the best case (C1), while achieving an average improvement of $24\%$ across all dataset-model configurations.

**Efficiency.** We evaluate the computational effort required to run each attack by reporting in Table 1 the mean runtime $s$ (in seconds), the mean number of queries $q$ issued to the model (in thousands), and the maximum VRAM used. Note that, while the runtime $s$ and the consumed VRAM may depend on the attack implementation, the number of queries $q$ counts the total number of forward and backward passes performed by the attack, thus providing a fairer evaluation of the attack complexity. In fact, some attacks perform more than 2000 queries even if $N = 1000$, i.e., they perform more than one forward and one backward pass per iteration (see, e.g., EAD and BB). Other attacks, instead, might use less than 2000 queries as they implement early stopping strategies. The results indicate that $\sigma\text{-zero}$ exhibits similar runtime performance when compared to the fastest algorithms FMN, PDPGD, and VFGA, while preserving higher effectiveness. In contrast, when compared against the BBadv attack, which competes in terms of $\tilde{\ell}_0$, our attack is much faster across all the dataset-model configurations, especially for Imagenet. For example, $\sigma\text{-zero}$ is 10 times faster than BBadv on C4 and 100 times faster on I3 on ImageNet. This confirms that $\sigma\text{-zero}$ establishes a better effectiveness-efficiency trade-off than that provided by state-of-the-art $\ell_0$-norm attacks.

**Reliability.** Complementary to Table 1, we present the robustness evaluation curves in Figure 2 for each attack on M2, C1, and I1. In Appendix B.3, we include similar curves for all other configurations. These curves go beyond the only median statistic and $ASR_k$, providing further evidence that $\sigma\text{-zero}$ achieves higher ASRs with smaller $\ell_0$-norm perturbations compared to the competing attacks. More importantly, the ASR of $\sigma\text{-zero}$ reaches almost always $100\%$ as the perturbation budget grows, meaning that its optimization only rarely fails to find an adversarial example. In Appendix B.1, we further demonstrate that even when the number of iterations is reduced to $N = 100$, $\sigma\text{-zero}$ consistently achieves an $ASR_\infty$ of $100\%$ across all models. This is not observed with other attacks, which often fail when using fewer iterations, thereby increasing the risk of overestimating adversarial robustness. These results reinforce our previous findings, confirming that $\sigma\text{-zero}$ can help mitigate the issue of overestimating adversarial robustness – a crucial aspect to foster scientific progress in defense developments and evaluations (Carlini et al., 2019; Pintor et al., 2022).

**Ablation Study.** In Table 2 we present an ablation study to evaluate the relevance of $\sigma\text{-zero}$'s components. Our findings indicate that all the non-trivial components in $\sigma\text{-zero}$ are essential for ensuring the effectiveness of the attack. Specifically, we observe that the $\ell_0$-norm approximation $\hat{\ell}_0$ (Eq. 7, line 3) leads the optimization algorithm to perturb all input features, albeit with small contributions. The projection operator (line 6) plays a crucial role by significantly decreasing the number of perturbed features, effectively removing the least significant contributions. Furthermore, gradient normalization (line 4) accelerates convergence, enhancing efficiency. Lastly, the adaptive projection operator (line 8) fine-tunes the results, reduces the number of perturbed features, and mitigates the dependency on hyperparameter choices. These results underline the importance of each component in $\sigma\text{-zero}$, highlighting their contributions to the overall performance of the attack.

Table 2: Ablation study on the $\sigma$-zero components integrated in Algorithm 1. Columns describe respectively: Gradient normalization factor (line 4); dynamic projection adjustment line 8; projection operator $\Pi_\tau$ (line 6); and the $\ell_0$ norm approximation $\hat{\ell}_0$ (line 3).

| Model | Normalization | Adaptive $\tau$ | Projection | $\hat{\ell}_0$ | $ASR_{10}$ | $ASR_{50}$ | ASR | $\tilde{\ell}_0$ |
|---|---|---|---|---|---|---|---|---|
| | ✓ | ✓ | ✓ | ✓ | 21.68 | **73.02** | 100.0 | **32** |
| | ✓ | | ✓ | ✓ | **21.89** | 71.66 | 100.0 | **32** |
| C10 | | ✓ | ✓ | ✓ | 16.81 | 39.76 | 100.0 | 65 |
| | | | ✓ | ✓ | 12.95 | 13.23 | 100.0 | 505 |
| | | | | ✓ | 12.95 | 12.95 | 100.0 | 3004 |
| | ✓ | | | ✓ | 12.95 | 12.95 | 100.0 | 3070 |
| | ✓ | ✓ | ✓ | ✓ | **37.27** | **82.92** | 100.0 | **20** |
| | ✓ | | ✓ | ✓ | 37.01 | 79.83 | 100.0 | 21 |
| C5 | | ✓ | ✓ | ✓ | 29.56 | 52.83 | 100.0 | 46 |
| | | | ✓ | ✓ | 25.46 | 32.84 | 100.0 | 144 |
| | | | | ✓ | 23.78 | 23.78 | 100.0 | 3064 |
| | ✓ | | | ✓ | 23.78 | 23.78 | 100.0 | 3068 |

Table 3: Fixed-budget comparison results with $N = 1000$ ($N = 2000$ for Sparse-RS) on MNIST and CIFAR-10 at budgets $k = 24, 50, 100$. Columns $q_{24}$ and $s_{24}$ show the average number of queries (in thousands) and the average execution time per sample (in seconds) at $k = 24$.

| Attack | M | $ASR_{24}$ | $ASR_{50}$ | $ASR_{100}$ | $q_{24}$ | $s_{24}$ | VRAM | M | $ASR_{24}$ | $ASR_{50}$ | $ASR_{100}$ | $q_{24}$ | $s_{24}$ | VRAM |
|---|---|---|---|---|---|---|---|---|---|---|---|---|---|---|
| | | | | | MNIST | | | | | | | | | |
| PGD-$\ell_0$ | | 73.99 | 99.90 | 100.0 | 2.00 | 0.09 | 0.04 | | 61.87 | 94.15 | 98.50 | 2.00 | 0.09 | 0.04 |
| Sparse-RS | | 79.54 | 96.35 | 99.79 | 0.83 | 0.21 | 0.04 | | **98.92** | 99.96 | 100.0 | 0.24 | 0.07 | 0.04 |
| sPGD$_p$ | M1 | 65.55 | 97.97 | 99.99 | 0.46 | 0.09 | 0.05 | M2 | 67.92 | 98.57 | 99.97 | 0.92 | 0.08 | 0.05 |
| sPGD$_u$ | | 82.79 | 99.65 | 100.0 | 0.09 | 0.08 | 0.05 | | 62.25 | 98.11 | 99.99 | 1.00 | 0.09 | 0.05 |
| $\sigma$-zero | | **83.71** | **99.98** | 100.0 | 0.43 | 0.02 | 0.06 | | 98.11 | **100.0** | 100.0 | 0.14 | 0.01 | 0.06 |
| | | | | | CIFAR-10 | | | | | | | | | |
| PGD-$\ell_0$ | | 38.18 | 59.67 | 87.19 | 2.00 | 0.78 | 1.90 | | 22.99 | 36.20 | 67.54 | 2.00 | 0.35 | 0.69 |
| Sparse-RS | | 72.51 | 86.59 | 94.28 | 0.77 | 0.36 | 1.95 | | 30.87 | 45.65 | 63.26 | 1.47 | 0.28 | 0.68 |
| sPGD$_p$ | C1 | 66.37 | 89.21 | 99.36 | 0.74 | 0.41 | 2.06 | C3 | 31.82 | 58.62 | 93.19 | 1.39 | 0.17 | 0.73 |
| sPGD$_u$ | | 66.33 | 91.07 | 99.75 | 0.72 | 0.41 | 2.06 | | 36.16 | 70.06 | 98.07 | 1.30 | 0.16 | 0.73 |
| $\sigma$-zero | | **77.08** | **95.33** | **99.95** | 0.65 | 0.29 | 2.07 | | **38.67** | **73.00** | 98.53 | 1.33 | 0.15 | 0.75 |
| PGD-$\ell_0$ | | 32.41 | 59.19 | 89.22 | 2.00 | 0.57 | 2.46 | | 34.35 | 44.99 | 68.61 | 2.00 | 0.35 | 0.70 |
| Sparse-RS | | 59.24 | 79.81 | 92.43 | 1.04 | 0.35 | 2.46 | | 49.35 | 63.01 | 76.51 | 1.11 | 0.37 | 0.68 |
| sPGD$_p$ | C2 | 58.91 | 88.15 | 99.42 | 0.89 | 0.39 | 2.57 | C4 | 50.41 | 75.86 | 97.52 | 1.02 | 0.18 | 0.73 |
| sPGD$_u$ | | 64.8 | 93.15 | 99.92 | 0.76 | 0.48 | 2.56 | | **55.89** | **84.64** | **99.56** | 0.91 | 0.19 | 0.73 |
| $\sigma$-zero | | **75.09** | **97.67** | **100.0** | 0.65 | 0.17 | 2.68 | | 55.69 | 82.72 | 99.07 | 0.94 | 0.11 | 0.75 |

**Comparison with Fixed-budget Attacks.** We complement our analysis by comparing $\sigma$-zero with three fixed-budget $\ell_0$-norm attacks, i.e., the $\ell_0$-norm Projected Gradient Descent (PGD-$\ell_0$) attack (Croce & Hein, 2019), the Sparse Random Search (Sparse-RS) attack (Croce et al., 2022),[7] and the Sparse-PGD attack (Zhong et al., 2024). For Sparse-PGD, we consider the implementation with sparse (sPGD$_p$) and with unprojected (sPGD$_u$) gradient. In contrast to minimum-norm attacks, fixed-budget attacks optimize adversarial examples within a given maximum perturbation budget $k$. For a fairer comparison, as done in fixed-budget approaches, we early stop the $\sigma$-zero optimization process as soon as an adversarial example with an $\ell_0$-norm perturbation smaller than $k$ is found. In these evaluations, we set $N = 1000$ for $\sigma$-zero, PGD-$\ell_0$, sPGD$_p$, and sPGD$_u$, while using $N = 2000$ for Sparse-RS. Therefore, when using $N = 1000$ steps for $\sigma$-zero (which amounts to performing 1000 forward and 1000 backward calls), we set $N = 2000$ steps for Sparse-RS (which

---

[7]Sparse-RS is a gradient-free (black-box) attack, which only requires query access to the target model. We consider it as an additional baseline in our experiments, but it should not be considered a direct competitor of gradient-based attacks, as it works under much stricter assumptions (i.e., no access to input gradients).

Table 4: Fixed-budget comparison results with $N = 1000$ ($N = 2000$ for Sparse-RS) on ImageNet at budgets $k = 100, 150$. See the caption of Table 3 for further details.

| Attack | M | $\text{ASR}_{100}$ | $\text{ASR}_{150}$ | $\mathbf{q}_{100}$ | $\mathbf{s}_{100}$ | VRAM | M | $\text{ASR}_{100}$ | $\text{ASR}_{150}$ | $\mathbf{q}_{100}$ | $\mathbf{s}_{100}$ | VRAM |
|---|---|---|---|---|---|---|---|---|---|---|---|---|
| ImageNet | | | | | | | | | | | | |
| Sparse-RS | | 89.3 | 91.5 | 0.39 | 0.32 | 1.29 | | 81.1 | 84.1 | 0.53 | 0.5 | 4.39 |
| $\text{sPGD}_p$ | I1 | 95.4 | 98.5 | 0.31 | 0.16 | 1.40 | I2 | 85.6 | 91.2 | 0.33 | 0.64 | 4.48 |
| $\text{sPGD}_u$ | | 93.6 | 97.8 | 0.33 | 0.12 | 1.40 | | 82.6 | 88.7 | 0.37 | 0.39 | 4.49 |
| $\sigma\text{-zero}$ | | **99.7** | **100.0** | 0.19 | 0.06 | 1.79 | | **94.7** | **97.1** | 0.15 | 0.17 | 4.90 |
| Sparse-RS | | 69.1 | 72.2 | 0.81 | 0.62 | 4.39 | | 45.9 | 47.4 | 1.17 | 1.12 | 5.72 |
| $\text{sPGD}_p$ | I3 | 85.4 | 93.4 | 0.32 | 0.55 | 4.49 | I4 | 66.3 | 74.9 | 0.73 | 1.39 | 5.84 |
| $\text{sPGD}_u$ | | 83.9 | 92.1 | 0.35 | 0.39 | 4.49 | | 66.0 | 76.0 | 0.72 | 1.01 | 5.84 |
| $\sigma\text{-zero}$ | | **97.7** | **99.6** | 0.34 | 0.37 | 4.90 | | **78.8** | **85.8** | 0.49 | 0.70 | 6.29 |

amounts to performing 2000 forward calls).[8]    Furthermore, to compute the ASR at different $k$ ($\text{ASR}_k$), we separately execute fixed-budget attacks for $k = 24, 50, 100$ features on MNIST and CIFAR-10, and with $k = 100, 150$ features on ImageNet (excluding PGD-$\ell_0$ due to computational demands), reporting only the maximum number of queries and execution time across all distinct runs. We report the average query usage at $k$ ($\mathbf{q}_k$) and the average execution time per sample at $k$ ($\mathbf{s}_k$). We report the execution time of $\mathbf{s}_k$ for the smaller $k$, as it requires, on average, more iterations due to the more challenging problem. The results, shown in Tables 3-4, confirm that $\sigma\text{-zero}$ outperforms competing approaches in 17 out of 18 configurations (see Appendix B.2 for additional results). Only against C4 the fixed-budget attack $\text{sPGD}_u$ slightly increases the ASR. The advantages of $\sigma\text{-zero}$ become even more evident when looking at the results on ImageNet, where, on average, it improves the $\text{ASR}_{100}$ of 9.6% across all models in Table 4. The results also indicate that early stopping enables $\sigma\text{-zero}$ to save a significant number of queries and runtime while preserving a high ASR. In Appendix B.2, we also report additional comparisons with $N = 2500$ and $N = 5000$, i.e. a more favorable scenario for the competing attacks, confirming that $\sigma\text{-zero}$ remains competitive even at higher budgets.

**Summary.** Our experiments show that $\sigma\text{-zero}$: (i) outperforms minimum-norm attacks by improving the success rate and decreasing the $\ell_0$ norm of the generated adversarial examples (see Table 1 and Appendix B.1); (ii) is significantly faster and scales easily to large datasets (see Table 1 and Appendix B.1); (iii) is robust to hyperparameter selection, not requiring sophisticated and time-consuming tuning (see Appendix A.2); (iv) does not require any adversarial initialization (see Table 1); (v) provides more reliable adversarial robustness evaluations, consistently achieving 100% ASRs (see Table 1, Figure 2, Appendix B.3); and (vi) remains competitive against fixed-budget attacks even when given the same query budget (Table 3-4).

## 4 RELATED WORK

Optimizing $\ell_0$-norm adversarial examples with gradient-based algorithms is challenging due to non-convex and non-differentiable constraints. We categorize them into two main groups: (i) multiple-norm attacks extended to $\ell_0$, and (ii) attacks specifically designed to optimize the $\ell_0$ norm.

**Multiple-norm Attacks Extended to $\ell_0$.** These attacks have been developed to work with multiple $\ell_p$ norms, including extensions for the $\ell_0$ norm. While they can find sparse perturbations, they often rely heavily on heuristics in this setting. Brendel et al. (2019a) initialize the attack from an adversarial example far away from the clean sample and optimizes the perturbation by following the decision boundary to get closer to the source sample. In general, the algorithm can be used for any $\ell_p$ norm, including $\ell_0$, but the individual optimization steps are very costly. Pintor et al. (2021) propose the FMN attack that does not require an initialization step and converges efficiently with lightweight gradient-descent steps. However, their approach was developed to generalize over $\ell_p$ norms, but does not make special adaptations to minimize the $\ell_0$ norm specifically. Matyasko & Chau (2021) use

---

[8] $N = 2000$ is suggested as a lower bound number of iterations to ensure the convergence of Sparse-RS by Croce et al. (2022). Additional results with $N = 5000/10000$ for Sparse-RS can be found in Appendix B.2.

relaxations of the $\ell_0$ norm (e.g., $\ell_{1/2}$) to promote sparsity. However, this scheme does not strictly minimize the $\ell_0$ norm, as the relaxation does not set the lowest components exactly to zero.

$\ell_0$-**specific Attacks.** Croce et al. (2022) introduced Sparse-RS, a random search-based attack that, unlike minimum-norm attacks, aims to find adversarial examples that are misclassified with high confidence within a fixed perturbation budget. On the same track we find Sparse-PGD (Zhong et al., 2024) and PGD-$\ell_0$ (Croce & Hein, 2019), white-box fixed-budget alternatives to Sparse-RS. Lastly, Césaire et al. (2021) induces folded Gaussian noise to selected input components, iteratively finding the set that achieves misclassification with minimal perturbation. However, it requires considerable memory to explore possible combinations and find an optimal solution, limiting its scalability.

Overall, current implementations of $\ell_0$-norm attacks present a crucial suboptimal trade-off between their success rate and efficiency, i.e., they are either accurate but slow (e.g., BB) or fast but inaccurate (e.g., FMN). This is also confirmed by a recent work that has benchmarked more than 100 gradient-based attacks (Cinà et al., 2025) on 9 additional robust models. In that open-source benchmark, $\sigma$-zero consistently and significantly outperformed all the existing implementations of competing $\ell_0$-norm attacks, establishing a performance very close to that of the empirical *oracle* (obtained by ensembling all the attacks tested). In summary, our attack combines the benefits of the two families of attack detailed above, i.e., effectiveness and efficiency, providing the state-of-the-art solution for adversarial robustness evaluations of DNNs when considering $\ell_0$-norm attacks.

## 5 CONCLUSIONS AND FUTURE WORK

In this work, we propose $\sigma$-zero, a novel attack aimed to find minimum $\ell_0$-norm adversarial examples, based on the following main technical contributions: (i) a differentiable approximation of the $\ell_0$ norm to define a novel, smooth objective that can be minimized via gradient descent; and (ii) an adaptive projection operator to enforce sparsity in the adversarial perturbation, by zeroing out the least relevant features in each iteration. $\sigma$-zero also leverages specific optimization tricks to stabilize and speed up the optimization. Our extensive experiments demonstrate that $\sigma$-zero consistently discovers more effective and reliable $\ell_0$-norm adversarial perturbations across all models and datasets while maintaining computational efficiency and robustness to hyperparameters choice. In conclusion, $\sigma$-zero emerges as a highly promising candidate to evaluate robustness against $\ell_0$-norm perturbations and promote the development of novel robust models against sparse attacks.

**Ethics Statement.** Based on our comprehensive analysis, we assert that there are no identifiable ethical considerations or foreseeable negative societal consequences that warrant specific attention within the limits of this study. This study will rather help improve the understanding of adversarial robustness of DNNs and identify potential ways to improve it.

**Reproducibility.** To ensure the reproducibility of our work, we have detailed the experimental setup in Section 3.1, where we describe the datasets, models, and attacks used, along with their respective sources. Additionally, we have provided our source code as part of the supplementary material, which will be made publicly available as open source upon acceptance.

### ACKNOWLEDGMENTS

This work has been partially supported by the project Sec4AI4Sec, under the EU's Horizon Europe Research and Innovation Programme (grant agreement no. 101120393); the project ELSA, under the EU's Horizon Europe Research and Innovation Programme (grant agreement no. 101070617); the EU—NGEU National Sustainable Mobility Center (CN00000023), Italian Ministry of University and Research (MUR) Decree n. 1033—17/06/2022 (Spoke 10); projects SERICS (PE00000014) and FAIR (PE0000013) under the MUR NRRP funded by the EU—NGEU; and by the German Federal Ministry of Education and Research under the grant AIgenCY (16KIS2012).

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
