# OpenReview forum: "$\sigma$-zero: Gradient-based Optimization of $\ell_0$-norm Adversarial Examples"
_ICLR.cc/2025/Conference — ICLR 2025 Poster_

### Official Review · Reviewer_xCu7 · 2024-10-29

**Soundness:** 3
**Presentation:** 3
**Contribution:** 3
**Rating:** 6
**Confidence:** 4

**Summary:**

The authors propose a $\ell_0$-norm attack, called sigma-zero, which leverages a differentiable approximation of the $\ell_0$ norm to facilitate gradient-based optimization. The attack can find minimum $\ell_0$-norm adversarial examples. The experiments show that sigma-zero exhibits good performance in different settings.

**Strengths:**

1. The paper is easy-to-follow

2. The method is simple but effective

3. The experiments are relatively comprehensive

**Weaknesses:**

1. Despite the effectiveness of sigma-zero on different models, the evaluation on $\ell_0$-robust models is missing. Please consider including the results of sAT / sTRADES [1], e.g., those trained on CIFAR-10, $k_{train}=6\times20$ in pixel space.

2. Due to the early-stopping mechanism widely adopted in various attacks, when the batch size is large, white-box attacks could run even faster than black-box attacks with the same iteration budget. Thus, I am still curious about the results of sigma-zero, sPGD and Sparse-RS with the same budget, e.g., N=10000. You can report the same metrics on a subset of representative models as in Table 13.

I will adjust my score if the authors can address my concerns.

[1]  Xuyang Zhong, Yixiao Huang, and Chen Liu. Towards efficient training and evaluation of robust models against l0 bounded adversarial perturbations.

**Questions:**

1. Typo on the title of Table 2: "Columns $q_{100}$ and $s_{100}$ ... at $k=100$", but $k$ is 24 in the below.

---

> ### Author Response · Authors · 2024-11-22
> **Response to xCu7**
>
> **Experiments on L0-Robust Models.** As requested, we expand our experiments to include the $\ell_0$ robust models sTRADES (denoted C11) and sAT (denoted with C12). Within the given time constraints, we decided to compare $\sigma$-zero with the most competitive attacks: FMN and BBadv for minimum-norm attacks, and Sparse-RS and Sparse-PGD for fixed-budget attacks. The camera-ready version will include results for the remaining (weaker) attacks, which we do not expect to significantly affect the following conclusions. Furthermore, for $\sigma$-zero we consider its default hyperparameters and a configuration using $\sigma=1$ (denoted with $\sigma$-zero*).
> We report our results in Table 6 (for minimum-norm attacks) and Table 10 (for fixed-budget attacks) in the revised manuscript. Overall, our findings indicate that $\ell_0$ gradient-based attacks (BB, Sparse-PGD, and $\sigma$-zero) require larger perturbations to break the model, demonstrating the model’s resistance to these attacks (as evidenced by poor ASR for fixed-budget attacks and large median $\tilde{\ell}_0$ values for minimum-norm attacks). Conversely, we observe that Sparse-RS and FMN perform better against these models.
> In particular, Sparse-RS, as a black-box attack, does not explicitly follow the gradient for minimizing $\ell_0$, while FMN optimizes adversarial examples in an $\ell_2$ direction (without using sparse projections on the gradient direction). These observations suggest that relying on the $\ell_0$  gradient direction (i.e. a sparsified gradient projection) when attacking these models can hinder the optimization of the attack, thus lowering its performance and leading to a false sense of security, while following a non-sparsified, dense gradient direction does not lead to the same problem.
> This observation is confirmed by the results obtained by $\sigma$-zero*, as using a larger value of the smoothing parameter $\sigma$ reduces sparsification of the gradient descent direction, overcoming the aforementioned issue. This adjustment effectively improves attack success rates against $\ell_0$ defenses (as shown in Tables 6 and 10, models C11 and C12). This flexibility highlights an advantage of $\sigma$-zero and suggests future strategies in which its smoothing parameter can be dynamically adjusted to overcome issues when attacking robust models against L0-norm attacks. We will add this discussion in the paper, and would like to thank the reviewer for requesting these additional experiments which unveiled a relevant aspect to further improve our strategy in this setting.
>
>
> **Experiments with N=10000** As requested, we include additional comparisons with fixed-budget attacks using 10,000 iterations. These experiments are performed on robust models from CIFAR-10 and ImageNet, with further results on the remaining models that will be included during camera-ready. The results, presented in  Table 15 of the appendix in the revised manuscript, demonstrate that $\sigma$-zero continues to deliver outstanding performance, surpassing the state of the art. The updated results are thus consistent with our earlier findings, further supporting the robustness and reliability of our conclusions.
>
>
> **Minor Issues** We thank the reviewer for the suggestion, we corrected the typo.
>
>
> We appreciate that you found our contributions both interesting and easy to follow. We hope that the additional results and responses we have provided will prompt the reviewer to consider increasing their score. We are working to complete the additional requested experiments in the coming days to further demonstrate the robustness and quality of our work.

---

> > ### Comment · Reviewer_xCu7 · 2024-11-22
> >
> > Thanks for the updates from the authors. My concerns have been addressed. I will accordingly adjust my score.

---

### Official Review · Reviewer_xveC · 2024-11-02

**Soundness:** 4
**Presentation:** 4
**Contribution:** 4
**Rating:** 8
**Confidence:** 5

**Summary:**

This paper proposes a $\ell_0$-norm adversarial attack that efficiently breaks various models trained on MNIST, CIFAR10, and ImageNet with 100% success rate. The proposed method is significantly faster than prior attacks at a high success rate. The proposed attack is composed of three main components:
1) A differentiable relaxation of the $\ell_0$ loss (Eq. 7).
2) An adaptive projection to project near-zero components to zero. The threshold for projection, $\tau$, is adapted dynamically such it is increased when the sample is adversarial and decreased otherwise.
3) Cosine annealing of the learning rate.

**Strengths:**

- The proposed method is highly effective against a diverse set of models on multiple datasets and all attack budgets (Table 1-3, Figure 2). Particularly, on ImageNet $\sigma$-zero improves the attack success rate at 100 by nearly 10% across all models (Table 3).
- Compared with BBadv that often achieves similar ASRk values, the proposed method is 10-200x faster.

**Weaknesses:**

- I recommend moving the ablations on the components of the proposed method (Table 5) to the main body of the paper. For completeness, consider adding the missing row with normalization/adaptive $\tau$ but without projection and dropping the column approximation as all rows rely on it.

**Questions:**

- Eq. 4 defines the regularized objective with a fixed regularization coefficient of $1/d$. How important is this normalization? Do you have any theories or ablations?
- BBadv is said to be slow, particularly it relies on adversarial initialization. Would $\sigma$-zero benefit in any way if it is also initialized the same way if we ignore the extra runtime?

Minor:
- Line 321: The sentence starts with “Despite BBadv does not suffer …” needs grammar correction, e.g., substituting “Despite” with “Although”.
- Consider annotating Figure 2 with subfigure captions.
- How is table 4 ordered? It seems to be random.

---

> ### Author Response · Authors · 2024-11-22
> **Response to xveC02**
>
> We thank the reviewer for their positive feedback on our work. We’re pleased that they found our contributions interesting and valuable. We hope the following changes will further strengthen our paper and encourage your support.
> Furthermore, we wish to inform the reviewer that we have extended our evaluation by integrating additional models to test $\sigma$-zero, further demonstrating its contributions to the state of the art. We now have a total of 22 models (originally they were 18) investigated in our experiments (i.e., 2 for MNIST, 12 for CIFAR-10, and 8 for ImageNet).
>
>
> **Move ablation study to the main paper.** In accordance with the reviewer, we have revised the manuscript to include and discuss the ablation study in the main paper. We would like to clarify that the "Adaptive $\tau$” column refers to the capacity of $\sigma$-zero to make the projection adaptive with respect to the thresholding value $\tau$. As such, it is not possible to have the "Adaptive \tau” setting without the “Projection” column, or it would have the same effect as simply using normalization. Additionally, the role of the "Approximation" column (now renamed as $\hat{\ell_0}$) is to emphasize that relying solely on this approximation leads to suboptimal results in terms of the number of perturbed features. This helps highlight the importance of the full approach and contributions of this paper for achieving better performance.
>
>
> **Normalization coefficient**. The scaling factor 1/d in Eq. 4 is used to keep the loss function within a 0 to 1 range, L0 can range from 0 to d, with d being the number of input features. From a design standpoint, it makes the loss function of our attack clearer to interpret by avoiding the usage of arbitrary soft-constraint weight terms.
>
>
> **Adversarial initialization**. Interesting question. During the development of $\sigma$-zero, we experimented with initializing the attack similarly to BBadv. However, we did not observe significant improvements in performance. This is likely because, as demonstrated in our experiments, $\sigma$-zero can reach adversarial examples within a few iterations and then use the remaining iterations to minimize the number of manipulated features. ​​However, Initial investigations suggest that a promising research direction would be to create a synergy between attacks at different stages of their optimization process. Some attacks perform better in certain phases, such as exploration versus exploitation. For instance, FMN is even faster at initializing adversarial examples to good local solutions. This opens the possibility of developing an ensemble approach where methods like FMN quickly produce locally good adversarial examples, which could then be further refined using $\sigma$-zero to further reduce their number of manipulations. We will discuss this as a potential future extension of this work.
>
>
> **Minor Issues**. We thank the reviewer for the helpful suggestions. We have corrected the typos in the paper and revised Table 4. The models are now listed in ascending order by their names in the Model column (e.g., from C1, C2, … to C10 for CIFAR-10).

---

> ### Comment · Reviewer_xveC · 2024-11-25
>
> I thank authors for their response to my questions. I do not have any outstanding concerns. I retain my positive rating and recommend acceptance.

---

### Official Review · Reviewer_L1s2 · 2024-11-02

**Soundness:** 2
**Presentation:** 2
**Contribution:** 2
**Rating:** 6
**Confidence:** 3

**Summary:**

The paper introduces σ-zero, a novel attack aimed at creating sparse adversarial examples under the ℓ0 norm constraint. By using a differentiable ℓ0 norm approximation and adaptive projection operator, the attack achieves high success rates across multiple benchmarks (MNIST, CIFAR-10, and ImageNet) and models. It outperforms existing sparse attacks in both efficiency and effectiveness.

**Strengths:**

1. **Innovative Approach**: σ-zero combines a unique differentiable approximation of the ℓ0 norm with adaptive projections, addressing challenges in sparse adversarial attack optimization with a gradient-based method.
2. **Robust Evaluation**: The experimental setup is thorough, involving extensive model types and datasets, and a range of comparison attacks. Evaluation metrics like success rate, median perturbation size, runtime, and query count provide a clear assessment of σ-zero's performance relative to existing methods.
3. **Efficiency and Robustness**: σ-zero demonstrates effective performance without the need for adversarial initialization or hyperparameter tuning, potentially broadening its applicability.

**Weaknesses:**

1. **Limited Innovation**
   The approach presented shows minimal originality and bears a strong resemblance to existing methods, with only slight modifications. These adjustments lack ingenuity, as they do not introduce new insights or significant advancements over prior work. The lack of a novel perspective limits the contribution of this study to the field.

2. **Lack of Theoretical Understanding and Analysis**
   A major shortcoming of this work is the absence of a robust theoretical foundation. The paper lacks in-depth analysis to demonstrate the underlying principles or broader significance of the method. Without a theoretical understanding, the impact of the work remains superficial, leaving readers without a clear sense of the method’s potential applicability or contribution to the advancement of adversarial robustness.

3. **Limited Application Scope**
   All experiments focus exclusively on white-box attacks, which limits the generalizability of the findings. While white-box evaluations are useful for testing model robustness, the absence of analysis in other contexts raises concerns about the method’s broader applicability. Expanding to include black-box or transfer-based attacks, for instance, could improve the study’s relevance and highlight potential real-world applications.

4. **Simplistic Model Choices in Experiments**
   Although the experiments involve several models, these are relatively simple in structure. The paper does not consider more complex architectures, such as ResNet-101 or models of similar or higher complexity, which are frequently used in evaluating ImageNet-related tasks. Given that larger and more sophisticated models are common in real-world applications, the study’s reliance on simpler architectures limits the robustness and scalability of its findings.


**Miner Comments:**
1. There is an indexing error in Table 2 that needs to be addressed.
2. Footnote 1 states, 'when the source point $x$ is already misclassified by $f $, the solution is simply $\delta^* = 0$. This is quite puzzling, as attacks are typically conducted on samples that can be correctly classified. If the attack is performed on samples that are already misclassified, I question the persuasiveness of the higher ASR observed in the experimental results.

**Questions:**

**Q1**: In the introduction, you mention that “evaluating adversarial robustness under these attacks could reveal weaknesses otherwise left untested with more conventional ℓ2- and ℓ∞-norm attacks.” Could you specify what types of model weaknesses can be exposed by ℓ0 attacks that may remain undetected under conventional ℓ2 and ℓ∞ attacks? Examples or specific scenarios would be helpful for understanding.

**Q2:** Have you considered any step size decay methods other than cosine annealing in your algorithm? Exploring more direct approaches might further reduce computational complexity.


**Q3:** Does your method enhance the improvement or understanding of attacks using other norms? Based on your assertion that 'ℓ0-norm attacks, which perturb only a minimal fraction of input values, can identify the most sensitive features affecting the model's decision-making,' could your method be integrated into attacks with other norms to avoid targeting these sensitive features, thereby reducing the visual damage of adversarial examples?


**Q4:** I appreciate your clarification that 'the goal of ℓ0-norm attacks is not to be indistinguishable to the human eye—a common misconception regarding adversarial examples—but rather to demonstrate whether and to what extent models can be deceived by altering only a few input values.' However, in the visualizations of adversarial examples provided in your appendix, the perturbations lack semantic information comprehensible to humans. How can we understand and apply the adversarial perturbations generated by your method in this context?

---

> ### Author Response · Authors · 2024-11-21
> **Reviewer L1s2 pt 1**
>
> We thank the reviewer for recognizing the value and contributions of our work.
>
>
> **Limited Innovation.** We respectfully disagree with the reviewer. The main value of our proposal lies in the combination of all the components considered, which are listed below and acknowledged by other reviewers. Specifically, contributions are:
>
> - *Objective*. The objective function in Eq. 6 is designed to induce an alternate optimization process between the logit loss and the L0-norm penalty, without introducing any hyperparameters. This is a novel contribution and avoids re-running the attack several times to tune the tradeoff hyperparameter, as done by EAD.
> - *L0-norm*. The approximation has never been used to implement sparse attacks, which normally use the L1 norm to deal with convex projections, yielding suboptimal solutions (see, e.g., EAD). Our L0 approximation is non-convex, but it enables $\sigma$-zero to find sparser perturbations.
> - *Adaptive projection*. Our non-naive adaptive projection operator is designed to promote sparsity in the solution, zeroing negligible components iteratively, and reducing the dependency on the choice of the initial step size.
> State-of-the-art performance. $\sigma$-zero significantly advances the state of the art for sparse attacks, as demonstrated in multiple experiments in the paper and the additional ones we provide in this rebuttal. As a consequence of that, $\sigma$-zero can work as a reliable tool for improving the robustness assessment of a target model against sparse attacks.
>
>
>
>
> **Lack of Theoretical Understanding and Analysis.** We agree with the reviewer that adding theoretical insights may improve our work, and will point out this limitation in the paper, even if the same issue affects many other attacks (if not all) published in top-tier venues (e.g., Carlini and Wagner, 2017b; Brendel et al., 2019b; Croce et al., 2021; Pintor et al., 2021; Zhong et al., 2024). A theoretical study of all these algorithms, investigating consistency, convergence rates, etc. would certainly help the community gaining a better understanding of their inner workings, but considering the effort and skills required, we believe it should be investigated as additional follow-up work. Nevertheless, we firmly believe that our paper still provides significant contributions that advance the state of the art. Our attack establishes new state-of-the-art performances for sparse attacks, as shown by our extensive experiments, both in terms of efficacy and efficiency. The other attacks are either fast but suboptimal (e.g., FMN, EAD) or accurate but extremely slow (e.g., BB, BBadv). $\sigma$-zero is the only L0-norm attack that is both fast and accurate, thereby enabling a scalable and effective robustness evaluation against sparse attacks. This makes it a worthy contribution for a venue such as ICLR.
>
>
> **Limited Application Scope.** We wish to clarify that our primary objective, as outlined in Section 2, is to present a minimum-norm attack for assessing model robustness in a white-box setting, a standard approach in the field (e.g., [Carlini and Wagner, 2017b; Brendel et al., 2019b; Croce et al., 2021; Pintor et al., 2021; Zhong et al., 2024]). We agree with the reviewer’s recommendation that analyzing transferability would be a valuable additional contribution to our work. However, we think that providing a detailed and fair transferability analysis of $\sigma$-zero will again require a large number of comparisons with the competing approaches, thereby requiring efforts that go beyond the scope of this work (i.e. the proposal of a novel L0 minimum-norm attack outperforming state-of-the-art, existing approaches). We will nevertheless acknowledge this as an interesting extension for future work, and will plan to study the transferability of sparse white-box gradient-based attacks more systematically, following e.g. the transferability analysis performed in [A].
>
>
> [A] Demontis et al. Why do adversarial attacks transfer? explaining transferability of evasion and poisoning attacks. USENIX security 2019.

---

> > ### Author Response · Authors · 2024-11-22
> > **Reviewer L1s2 pt 2**
> >
> > **Simplistic Model Choices in Experiments.**  We have extended the experimental investigation for the ImageNet dataset to include additional larger models. Specifically, we now include robust models from Peng et al.[A] (\~87.8 million parameters, denoted with I7) and Mo et al.[B] (\~266 million parameters, denoted with I8), bigger than ResNet101 (around 44.5 million parameters).
> > Due to the limited time, we focused on comparing $\sigma$-zero with the most competitive and representative attacks: FMN and BBadv for minimum-norm attacks, and Sparse-RS and Sparse-PGD for fixed-budget attacks. The remaining attacks will be included in the camera-ready version of the paper; however, we expect these will not significantly impact the overall findings, as none of them have outperformed $\sigma$-zero across the other 20 tested models.
> > The results, reported in Tables 6 and 11 in the revised Appendix, further remark on the advances that $\sigma$-zero brings to the state of the art.  We now have a collection of 22 total models investigated in our experimentation (i.e.,  2 for MNIST, 12 for CIFAR-10, and 8 for ImageNet).
> >
> >
> > [A] Peng et al. Robust principles: Architectural design principles for adversarially robust CNNs. BMVC 2023.
> >
> > [B] Mo et al. When adversarial training meets vision transformers: Recipes from training to architecture. NeurIPS 2022.
> >
> >
> > **Response to Q1**. Sparse $\ell_0$-norm attacks, like $\sigma$-zero, find adversarial perturbations that are substantially different from those found by dense $\ell_2$- and $\ell_\infty$-norm attacks. These attacks indeed perturb only a minimal number of input values, making them particularly effective at uncovering the model's reliance on specific, highly influential input values—whether spurious or critical. This sparse perturbation approach also serves as a counterfactual analysis tool, offering insights into how minimal input changes can alter model predictions [A, B, C]. Moreover, sparse perturbations may be relevant beyond image-based tasks, e.g., to implement meaningful attacks in the context of malware detection, where attackers may wish to minimize the number of features they can modify [D], e.g., corresponding to the injection or removal of system calls or actions.
> >
> >
> > [A] Freiesleben, Timo. The intriguing relation between counterfactual explanations and adversarial examples. 2022.
> >
> > [B] Dand et al. Multi-objective counterfactual explanations. 2020.
> >
> > [C] Wachter et al. Counterfactual explanations without opening the black box. 2017.
> >
> > [D] Cara, et al. On the feasibility of adversarial sample creation using the android system api. 2020.
> >
> >
> >
> >
> > **Response to Q2** Step-size annealing is widely used in attacks that normalize the input gradient in each step, to iteratively reduce the size of the perturbation update and speed up convergence, while also helping the algorithm to find better minima (as reducing the step size improves the exploitation phase of the optimization process) [I, II, III]. During the development of $\sigma$-zero we realized that gradient normalization and cosine annealing empirically led to finding better solutions, coherently with the findings in prior work. We have revised the paper to better explain the role and definition of the cosine-annealing strategy.
> >
> >
> > [I] Pintor et al. Fast minimum-norm adversarial attacks through adaptive norm constraints. NeurIPS, 2021.
> >
> > [II] Rony et al. Decoupling direction and norm for efficient gradient-based l2 adversarial attacks and
> > defenses. IEEE/CVF CVPR 2018.
> >
> > [III] Hoki et al. Generating transferable adversarial examples for speech classification. Pattern Recognition 2023.
> >
> >
> > **Response to Q3 and Q4** The sparse perturbations generated are not intended to be semantically interpretable to humans but instead highlight which minimal input changes can deceive the model. This information is valuable for understanding dependencies on specific input features and then assessing whether tested models align with human reasoning or reveal spurious correlations, such as reliance on background information. Furthermore, these insights can be applied to refine model training, ensuring robustness against attacks targeting such features (e.g., [Zhong et al., 2024)]. We thus believe that $\sigma$-zero provides a valuable, additional asset towards improving the current understanding of the robustness and inner workings of state-of-the-art, deep learning models.

---

> > > ### Author Response · Authors · 2024-11-22
> > > **Reviewer L1s2 pt 3**
> > >
> > > **Minor** We have addressed the minor writing issues as suggested. Regarding footnote 1, we clarified that if a sample is already misclassified, the optimal solution for a minimum-norm adversarial example is $\delta = 0$. Additionally, following the classical approach for adversarial evaluation (e.g., [Carlini and Wagner, 2017b; Brendel et al., 2019b; Pintor et al., 2021]), all attacks begin with the same ASR, equal to 1 minus the model’s accuracy, ensuring evaluation on the same set of samples across experiments. Furthermore, we provide full fairness and transparency about that in the paper. Readers can indeed verify this in Figure 2 (right), where all attacks start from the same ASR (~30%). There is no performance bias, and results are directly comparable, as scaling factors are consistent across models.
> > >
> > >
> > > We thank the reviewer for their constructive feedback and will incorporate these updates into the final version of the manuscript.

---

> > > > ### Author Response · Authors · 2024-11-26
> > > > **Follow-up on rebuttal**
> > > >
> > > > Dear Reviewer,
> > > >
> > > > Thank you once again for your detailed comments and suggestions. As the rebuttal period is nearing its end, we would greatly appreciate your feedback on whether our responses have addressed your concerns. If our responses and experiments have effectively addressed your points, we would be grateful if you could consider revising your scores accordingly. We are also happy to engage in further discussions if needed.
> > > >
> > > > Best regards,

---

> > > > > ### Comment · Reviewer_L1s2 · 2024-11-27
> > > > >
> > > > > Thank you for the updates and thoughtful responses from the authors. My concerns have been partially addressed, and I will adjust my score accordingly. I’ve learned a lot from this exchange and look forward to the opportunity to discuss some of these details further in the future.

---

### Official Review · Reviewer_fo9d · 2024-11-03

**Soundness:** 3
**Presentation:** 3
**Contribution:** 3
**Rating:** 6
**Confidence:** 3

**Summary:**

The paper proposes a novel $\ell_0$-norm attack by leveraging a differentiable approximation of the $\ell_0$-norm constraint. The approach can be applied to finding both minimum $\ell_0$-norm and fixed budget adversarial examples. The authors conduct extensive evaluations on diverse datasets (e.g. CIFAR-10, ImageNet) and models. The results show that the proposed attack is effective and efficient in reducing the number of queries and memory usage.

**Strengths:**

- The problem is well-motivated as $\ell_0$-norm attacks are less studied due to the non-convexity nature.
- The paper is easy to follow and well-organized
- The evaluations are comprehensive and indeed show that the proposed algorithm is more effective and efficient.

**Weaknesses:**

- I appreciate the authors' efforts in evaluating the method. However, I was a bit confused by the inconsistency in some parts of the evaluations. For example, the memory usage is shown for minimum-norm attacks while it's missing in the bounded-norm attack results; The mean runtime for bounded attacks is shown in Tables 2 and 3 while it's missing in Tables 10 - 12; In Tables 10 - 12, the budget level that the number of queries corresponds to is also missing.

**Questions:**

- To enable a clearer comparison between different attacks, could the authors provide ASR under varying iterations and perturbation budgets for fixed-budget attacks, similar to Figure 2? Showing results on one model would suffice and help better understand the performance under different conditions.
- The transferability of the proposed attack across different model and architectures is unclear. Would it be possible to evaluate the method on a different model/architecture than the one used for generating adversarial examples? Such an evaluation would strengthen the applicability of the attack across varied settings.
- In the minimum-norm attacks, ASR of the three datasets are reported on the same budgets: $k = 24, 50,\infty$. However, in the fixed-budget comparisons, the results of ImageNet are reported at budgets $k = 100, 150$ while the other two datasets are evaluated at $24, 50, 100$. Could the authors clarify the reasoning behind this difference?
- Minor writing issues
  - The bounded-norm attacks are not cited at the first appearance in Section 3.1
  - In Table 2’s caption, $q_{100}$ and $s_{100}$ are mentioned, but it seems these should be $q_{24}$ and $s_{24}$, based on the table’s content.

---

> ### Author Response · Authors · 2024-11-21
> **Response to Reviewer fo9d**
>
> We thank the reviewer for recognizing the value and contributions of our work.
>
>
> **Clarifications on memory usage and runtime.** We revised the paper to include memory usage and mean runtime results for the fixed-budget attacks. We also changed the column names to specify that the number of queries shown in Tables 10-15 corresponds to the smallest budget k, as this setting typically results in the highest average query consumption.
>
>
> **Clearer comparisons with fixed-budget attacks.**  To address the reviewer's request, we provide additional robustness evaluation curves in Figure 5 (in the new Appendix) for fixed-budget attacks across 8 different values of k on robust model C3. In this evaluation, we run fixed-budget attacks with $N=1000$ (and $N=5000$) iterations for each $k$, and the results consistently demonstrate that $\sigma$-zero outperforms existing fixed-budget attacks at every perturbation size $k$. We will provide the robustness evaluation curves related to the other models for the camera ready. We hope this addition offers a clearer understanding of the performance of $\sigma$-zero against fixed-budget attacks, confirming that our algorithm significantly outperforms them both in efficacy and efficiency.
>
>
> **Transferability to other architectures** We agree with the reviewer’s recommendation that analyzing transferability would be a valuable additional contribution to our work. However, we think that providing a detailed and fair transferability analysis of $\sigma$-zero will again require a large number of comparisons with the competing approaches, thereby requiring efforts that go beyond the scope of this work (i.e. the proposal of a novel L0 minimum-norm attack outperforming state-of-the-art, existing approaches). We will nevertheless acknowledge this as an interesting extension for future work, and will plan to study the transferability of sparse white-box gradient-based attacks more systematically, following e.g. the transferability analysis performed in [A].
>
>
> [A] Demontis et al. Why do adversarial attacks transfer? explaining transferability of evasion and poisoning attacks. USENIX security 2019.
>
>
> **Budget Inconsistencies** Thank you for pointing this out. The difference in the reported budgets stems from the fact that ImageNet has significantly more features than the other two datasets, thus making the optimization problem more challenging to solve as the adversarial perturbations must navigate a higher-dimensional input space while maintaining effectiveness. Therefore,  considering the perturbation levels used in previous work (e.g., Croce et al.), we set higher perturbation budgets for ImageNet to account for the increased complexity. In the final camera-ready version, we plan to include additional values of k for ImageNet, similar to the approach used for CIFAR-10, to provide a more consistent comparison across datasets.
>
>
> **Minor Writing Issues** Lastly, we have also corrected the minor writing issues as suggested.
>
>
> We will incorporate these updates in the final version of the manuscript, and hope that the reviewer will reconsider their evaluation.

---

> > ### Author Response · Authors · 2024-11-26
> > **Follow-up on rebuttal**
> >
> > Dear Reviewer,
> >
> > Thank you once again for your detailed comments and suggestions. As the rebuttal period is nearing its end, we would greatly appreciate your feedback on whether our responses have addressed your concerns. If our responses and experiments have effectively addressed your points, we would be grateful if you could consider revising your scores accordingly. We are also happy to engage in further discussions if needed.
> >
> > Best regards,

---

> > > ### Comment · Reviewer_fo9d · 2024-11-26
> > >
> > > Thanks for the authors' reply. All of my questions have been resolved, and I'll keep my positive rating.

---

### Meta-Review · Area_Chair_ANaW · 2024-12-20

**Metareview:**

The submission evaluates adversarial robustness of networks under \ell_0 norm attacks.

+ Paper is well-written.
+ Evaluations are comprehensive.

- Some technical details required clarity.
- Somewhat limited novelty and scope.

**Additional Comments On Reviewer Discussion:**

Some issues were raised in the initial reviews, including
- inconsistencies in evaluation
- lack of theoretical analysis
- use of simplistic models in evaluation
These issues were clarified in the rebuttal and should be included in the final version. All reviewers recommend acceptance.

---

### Decision · Program_Chairs · 2025-01-22

Accept (Poster)